# Non-equilibrium dynamics of dipolar polarons

Artem G. Volosniev[1]*, Giacomo Bighin[1,2], Luis Santos[3], Luis A. Peña Ardila[3,4†]

**1** Institute of Science and Technology Austria (ISTA), Am Campus 1, 3400 Klosterneuburg, Austria
**2** Institut für Theoretische Physik, Universität Heidelberg, D-69120 Heidelberg, Germany
**3** Institut für Theoretische Physik, Leibniz Universität Hannover, Germany
**4** School of Science and Technology, Physics Division, University of Camerino, Via Madonna delle Carceri, 9B - 62032 (MC), Italy.
* artem.volosniev@ist.ac.at
† luis.penaardila@unicam.it

December 12, 2023

## Abstract

We study the out-of-equilibrium quantum dynamics of dipolar polarons, i.e., impurities immersed in a dipolar Bose-Einstein condensate, after a quench of the impurity-boson interaction. We show that the dipolar nature of the condensate and of the impurity results in anisotropic relaxation dynamics, in particular, anisotropic dressing of the polaron. More relevantly for cold-atom setups, quench dynamics is strongly affected by the interplay between dipolar anisotropy and trap geometry. Our findings pave the way for simulating impurities in anisotropic media utilizing experiments with dipolar mixtures.

# 1  Introduction

Non-equilibrium dynamics is one of the most challenging and complex branches of quantum many-body physics [1–3]. One of its central questions is how a locally excited system reaches an equilibrium state. A paradigmatic testbed for studies of this question is a mobile impurity interacting with a quantum environment whose steady-state solution – a polaron [4] – provides a physical intuition behind low-energy transport in semiconductors [5], colossal magnetoresistance [6], non-equilibrium phenomena such as quantum heat transport [7] as well as polaron pairs in semiconducting organic polymers [8]. Unfortunately, it appears challenging to experimentally resolve time dynamics of the polaron in solid state systems, where natural timescales are 'fast', often given by femtoseconds (for electrons) and picoseconds (for lattice).

To study time evolution of systems with impurities, one can use ultracold quantum gases instead [9, 10], where, depending on the nature of the particles in the bath, impurities can form Fermi [11–14] or Bose polarons [15–18]; the latter are in the focus of our study. By rapidly populating the impurity state and probing its quench dynamics with radio frequency (RF) pulses, current cold-atom experiments can unveil the relevant timescales of Bose-polaron formation and provide insight into (i) how many-body impurity-bath correlations are built from the microscopic level and (ii) how an initially excited bath thermalize due to dephasing of phonons. In particular, a combination of a RF pulse that coherently populates the impurity state and Ramsey interferometry, which tracks down the quantum evolution of the impurity coupled coherently to the BEC, allows one to measure a time-dependent Green's function, which contains information about non-equilibrium dynamics of the impurity [9, 10]. Its imaginary part gives the contrast, namely the overlap between the initial and the final states. It is one of the key observables studied in this work.

Complexity of quench dynamics of impurities stems in particular from the interplay between few- and many-body correlations and the corresponding timescales. Initial dynamics involves a local excitation of the system, which implies participation of phonons with high momenta that can resolve few-body physics. After some time, the dynamics is mainly driven by propagation of low-energy many-body excitations. In a three-dimensional Bose gas, these limiting cases and the transition between them can be studied using several robust theoretical approaches based upon variational coherent ansatz [19–21], T-matrix approximation [22], Langevin equation [23], master equation [24, 25], the many-body generalization of Weisskopf-Wigner theory [26], dynamical variational ansatz [27], exact results [28]. [For impurities in a Fermi gas, see, for example, Refs. [29–31].] However, experimental and theoretical work on non-equilibrium polaron dynamics is so far limited to ultracold gases with short-range interactions. The connection between the far-from-equilibrium dynamics and the equilibrium polaron state remains an open problem for more complicated interactions.

If the range of the potential is of the same order as the interparticle distance, the static properties of impurities are distinctly different from those for short-range interactions. For example, Rydberg [32, 33] and ionic impurities [34, 35][1] are dressed by many atoms and form many-body

---

[1]In the Rydberg case, the effective impurity-atom potential contains both $s-$ and $p-$ wave scattering terms. For a charged impurity, the potential has an isotropic $1/r^4$ tail.

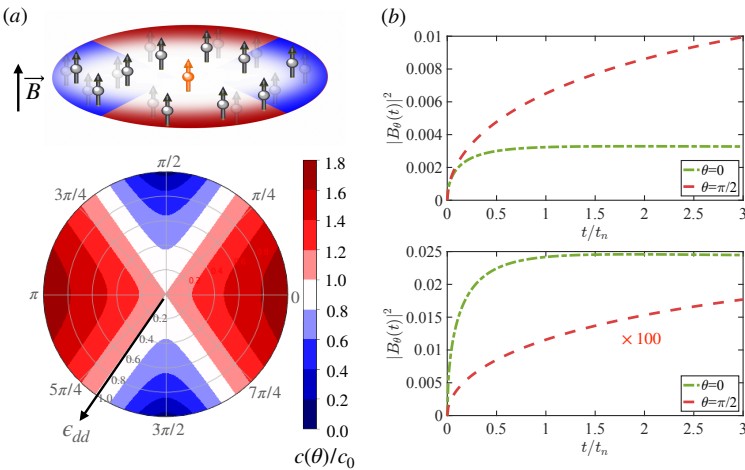

Figure 1: **(a)** Speed of sound $c(\theta)$ for different strengths of dipolar coupling, $\epsilon_{dd}$, depending on the relative direction between the phonon's momentum and the direction of the magnetic field, $\vec{B}$, which fixes the orientation of the dipoles; $c_0$ is the speed of sound for a non-dipolar system. **(b)** Population of phonons excited by the impurity in two directions with respect to the impurity. For $\theta = 0$ ($\theta = \pi/2$), the momentum of phonons is parallel (perpendicular) to the direction of the dipoles. The top plot is for a non-dipolar impurity ($d_2 = 0$). The bottom plot is for a dipolar impurity with $d_2 = 130a_0$ ($a_0$ is the Bohr radius). We consider a homogeneous gas with $na_{11}^3 = 5 \times 10^{-5}$ and $a_{11} = a_{12} = 150a_0$ so that $\epsilon_{dd} \simeq 0.87$. $\epsilon_{dd}$ is the parameter that characterizes the dipolar character of the bath, see the text for details.

bound states for strong interactions, which are beyond the so-called Fröhlich Hamiltonian [36,37]. Another scenario is provided by an impurity in a dipolar gas [38,39] where both the atom-atom and impurity-atom potentials are characterized by a long-range and anisotropic dipole-dipole interaction (DDI). The experimentally relevant interaction regimes of this system can be described using the Fröhlich Hamiltonian, making the dipolar polaron model an ideal platform for simplified theoretical studies of non-equilibrium dynamics in beyond-contact-interaction polaron models.

Motivated by the realization of bosonic mixtures of non-magnetic and magnetic atoms such as Er-Li [40] and Yb-Er [41] as well as of magnetic atoms such as Er and Dy [42–44], in this work, we study non-equilibrium physics of dipolar polarons, i.e., impurities (non-dipolar or dipolar) immersed in a Bose bath with dominant DDI. In stark contrast to systems ruled by short-range interactions, the dispersion relation in dipolar systems displays an anisotropic behavior at low energies [45,46], supporting, for certain parameters, exotic states of matter such as quantum droplets and supersolids [47,48]. As we shall demonstrate, the anisotropy of dipolar interactions also plays an important role in our study.

To investigate the quench dynamics of the impurity, we rely on a variational ansatz, see Sec. 3. We establish closed-form expressions for the parameters that enter this ansatz (see appendicies) and calculate the Ramsey contrast as the overlap between the non-interacting and the interacting states, see Sec. 4. At long times, the contrast is determined by the steady-state properties of the system, such as the polaron energy and the residue, and we start Sec. 4 by discussing these quantities. Then, we unveil the relevant timescales of the polaron formation and demonstrate that they contain information about the dipolar nature of the condensate, in particular, how far the system is from the collapse, see Sec. 4.3. Note that the Ramsey contrast is an integral quantity and cannot be used to study anisotropic nature of dipole-dipole interactions in a direct manner. Therefore, we introduce a theoretical construct in Sec. 4.4 that allows us to perform such a study and to show that

time evolution of the system is highly anisotropic. Finally, we discuss an experimentally relevant trapped case in Sec. 4.5.

For a non-dipolar impurity, our findings can be observed experimentally using well-established interferometric techniques such as many-body Ramsey spectroscopy see, e.g., [9, 10] or by driving Rabi oscillations between weakly and strongly interacting states [49]. In contrast, for the fully dipolar case, further decoherence processes due to fast dipolar relaxation may play a role. We leave an investigation of these effects to future studies.

## 2 System

We consider a highly imbalanced dipolar binary mixture at zero temperature. We first focus on a uniform three-dimensional system, see Sec. 4.5 for a discussion of an experimentally relevant trapped gas. The problem is cast in the framework of a single impurity problem interacting with a dipolar condensate. The host bath is labeled by $|1\rangle$ with density $n$ and the impurity is labeled by $|2\rangle$. For equal mass components, $m_1 = m_2 = m$, the out-of-equilibrium dynamics of the impurity obeys the Fröhlich Hamiltonian,

$$\hat{\mathcal{H}} = \sum_{\mathbf{k}} \omega_{\mathbf{k}} \hat{b}_{\mathbf{k}}^{\dagger} \hat{b}_{\mathbf{k}} + \frac{\hat{\mathbf{p}}^2}{2m} + nV_{12}(\mathbf{k}=0) + \frac{\sqrt{n}}{\sqrt{V}} \sum_{\mathbf{k}} V_{12}(\mathbf{k}) W_{\mathbf{k}} e^{-i\mathbf{k}\cdot\hat{\mathbf{r}}} \left( \hat{b}_{\mathbf{k}} + \hat{b}_{-\mathbf{k}}^{\dagger} \right), \quad (1)$$

which describes the scattering between an impurity with momentum operator $\hat{\mathbf{p}}$ and a Bogoliubov bosonic excitation of energy $\hbar\mathbf{k}$. Here, $\hat{b}_{\mathbf{k}}(\hat{b}_{\mathbf{k}}^{\dagger})$ annihilates (creates) an excitation with energy $\omega_{\mathbf{k}} = \sqrt{\epsilon_{\mathbf{k}}(\epsilon_{\mathbf{k}} + 2nV_{11}(\mathbf{k}))}$ where $\epsilon_{\mathbf{k}} = \hbar^2\mathbf{k}^2/2m$ is the free energy dispersion, and $W_{\mathbf{k}} = \sqrt{\epsilon_{\mathbf{k}}/\omega_{\mathbf{k}}}$; $\hat{\mathbf{r}}$ is the position operator for the impurity. The boson-boson ($V_{11}$) and impurity-boson ($V_{12}$) potentials are given by $V_{ij}(\mathbf{k}) = g_{ij} + g_{ij}^{d}\left(3k_z^2/k^2 - 1\right)$. The first term of this expression describes the short-range physics characterized by the coupling strength $g_{ij}$, which within the Born approximation reads $g_{ij} = 4\pi\hbar^2 a_{ij}/m$, where $a_{ij}$ is the $s$-wave scattering length. The second part contains the long-range DDI potential, which is anisotropic. Its strength is determined by the dipolar coupling strength $g_{ij}^{d} = \mu_0\mu_i\mu_j/3 = 4\pi\hbar^2\sqrt{d_i d_j}/m$, where $\mu_0$ is the vacuum permeability, and $\mu_{i(j)}$ is the magnetic dipolar moment related to the dipolar length $d_{i(j)}$.

The anisotropy of the DDI implies that the velocity of Bogoliubov excitations depends on the direction [50]. Figure 1 (a) depicts the anisotropic speed of sound $c(\theta)/c_0 = \sqrt{1 + \epsilon_{dd}(3\cos^2\theta - 1)}$, where $\epsilon_{dd} = d_1/a_{11}$ is the ratio between the dipolar and scattering lengths of the bath, and $c_0 = \sqrt{g_{11}n/m}$ is the sound velocity for the isotropic non-dipolar case ($\epsilon_{dd} = 0$). In the absence of dipolar interactions, polaron formation time, which marks the onset of the final stage of the impurity dynamics, is given by $\xi/c_0$ [25], where $\xi = \hbar/(\sqrt{2}c_0)$ is the coherence length of a non-dipolar condensate. If we naively define a direction-dependent timescale as $\hbar^2/(\sqrt{2}c(\theta)^2)$ (see App. D for a derivation of this timescale), then we shall expect slower dynamics in the side-to-side ($\theta = \pi/2$) direction, which features the softest excitations in the system [50], than in the head-to-tail ($\theta = 0$) one. This expectation is directly confirmed by our calculations, see Fig. 1 (b). The number of excited bosons is also direction-dependent, as we discuss in Sec. 4.4. Note that excitations with momentum perpendicular to the direction of dipoles ($\theta = \pi/2$) may be unstable [50], i.e., the speed of sound may become imaginary, which leaves a trace in the dynamics of the impurity, as we show in Sec. 4.3.

To conclude the problem formulation, let us motivate the use of the Fröhlich approximation, which is expected to be accurate only for weak interactions. As our focus is on the dynamics and decoherence of the impurity due to the interplay between the short and long-range anisotropic physics,

we shall only explore systems with $a_{ij} \simeq d_i$, i.e., the $s$-wave scattering length that determines the short-range physics is comparable to the dipolar length of magnetic atoms. For Dy, $d_1 \simeq 130a_0$, which for experimentally relevant densities implies a small gas parameter ($na_1^3 \ll 1$), allowing us to use the Fröhlich Hamiltonian as long as $\epsilon_{dd} < 1$, see also App. A.

# 3 Variational ansatz

To investigate the out-of-equilibrium dynamics of the system, we employ a coherent-state variational ansatz [19–21, 51] for the wave function $|\psi(t)\rangle = e^{-i\phi(t)} e^{\sum_{\mathbf{k}} \beta_{\mathbf{k}}(t) \hat{b}_{\mathbf{k}}^\dagger - \beta_{\mathbf{k}}^*(t) \hat{b}_{\mathbf{k}}} |0\rangle$. Here, $|0\rangle = |0_{\mathbf{k}}, \mathbf{P}\rangle$ is the state of the system at $t = 0$, which consists of an impurity with momentum $\mathbf{P}$ and a phononic vacuum[2]. The parameters $\beta_{\mathbf{k}}(t)$ and $\phi(t)$ are calculated using the standard principles of time-dependent variational theory, which minimizes the action functional $\int dt \left\langle \psi(t) \left| i\hbar\partial_t - \hat{\mathcal{H}} \right| \psi(t) \right\rangle$ [52] with the Hamiltonian after the Lee-Low-Pines transformation [53]: $\hat{\mathcal{H}} = \hat{S}^{-1} \hat{\mathcal{H}} \hat{S}$, here $\hat{S} = \exp\left( i\hat{\mathbf{r}} \cdot (\mathbf{P} - \hat{\mathbf{P}}_{\mathrm{B}}) \right)$ (see App. B). The corresponding equations for $\beta_{\mathbf{k}}(t)$ and $\phi(t)$ read (see App. C)

$$i\hbar\dot{\beta}_{\mathbf{k}} = \frac{\sqrt{n}}{\sqrt{V}} V_{12}(\mathbf{k}) W_{\mathbf{k}} + \Omega_{\mathbf{k}} \beta_{\mathbf{k}},$$

$$\hbar\dot{\phi}(t) = V_{12}(\mathbf{0}) n + \frac{\mathbf{P}^2 - \mathbf{P}_{\mathrm{B}}^2}{2m} + \frac{\sqrt{n}}{\sqrt{V}} \sum_{\mathbf{k}} W_{\mathbf{k}} V_{12}(\mathbf{k}) \mathfrak{Re}[\beta_{\mathbf{k}}],$$

(2)

where $\Omega_{\mathbf{k}} = \omega_{\mathbf{k}} + \epsilon_{\mathbf{k}} - \hbar\mathbf{k}(\mathbf{P} - \mathbf{P}_{\mathrm{B}})/m$. The momentum of the bosons, $\mathbf{P}_{\mathrm{B}} = \sum_{\mathbf{k}} \hbar\mathbf{k} |\beta_{\mathbf{k}}|^2$, will be neglected in our calculations (see below). In addition, we note that Eq. (2) should be regularized (see the discussion in App. B).

# 4 Results

## 4.1 Steady state

First, we study static properties of the system with $\mathbf{P} = 0$, by imposing in Eq. (2) the saddle-point condition, i.e., $\dot{\beta}_{\mathbf{k}} = 0$. The corresponding solution is $\beta_{\mathbf{k}} = -\sqrt{n} V_{12}(\mathbf{k}) W_{\mathbf{k}} / (\sqrt{V} \Omega_{\mathbf{k}})$. Note that $\beta_{\mathbf{k}}$ is proportional to $V_{12}(\mathbf{k})$, thus, $\mathbf{P}_{\mathrm{B}}$ enters in the next-to-leading order (hierarchy is determined by powers of $V_{12}$). Therefore, $\mathbf{P}_{\mathrm{B}}$ should not be considered when investigating the leading order effects, validating the neglect of $\mathbf{P}_{\mathrm{B}}$ in our calculations.

Let us now consider the equation of motion for the phase $\phi$. We identify the polaron energy as $E_0 = \hbar\dot{\phi}(t \to \infty)$, leading to $E_0 = V_{12}^\Lambda n - \frac{n}{V} \sum_{\mathbf{k}}^\Lambda \frac{(V_{12}(\mathbf{k}) W_{\mathbf{k}})^2}{\omega_{\mathbf{k}} + \epsilon_{\mathbf{k}}}$. The first term here corresponds to the mean-field result, whereas the second one includes quantum fluctuations at the level of second-order perturbation theory. For the considered parameters, the second term is about

---

[2] As we show below $\sum_k |\beta_k|^2 \to 0$ for weak impurity-boson interactions (i.e., $V_{12} \to 0$), which implies that $e^{\sum_{\mathbf{k}} \beta_{\mathbf{k}}(t) \hat{b}_{\mathbf{k}}^\dagger - \beta_{\mathbf{k}}^*(t) \hat{b}_{\mathbf{k}}} \simeq 1 + \sum_{\mathbf{k}} \left[ \beta_{\mathbf{k}}(t) \hat{b}_{\mathbf{k}}^\dagger - \beta_{\mathbf{k}}^*(t) \hat{b}_{\mathbf{k}} \right]$. Therefore, the coherent-state variational ansatz in our calculations corresponds (approximately) to an ansatz based upon a single-phonon excitation. This means that our steady-state results should be accurate at the level of second-order perturbation theory where perturbation is given by $V_{12}$, see also App. B where we calculate the energy perturbatively. Furthermore, as we focus on the Fröhlich Hamiltonian and our ansatz describes (approximately) a one-phonon excitation, relaxation in quench dynamics should occur via dephasing of the initially populated continuum of energy states (cf. Ref. [28]).

10% of the mean-field result, see App. B. Note that we have employed the regularized potential $V_{12}^{(\Lambda)} = \frac{4\pi\hbar^2}{m}\left[a_{12} + d_1 d_2 \frac{8}{5\pi}\Lambda + \frac{2a_{12}^2}{\pi}\Lambda\right]$, where $\Lambda$ is a high-energy cut-off parameter. For a more detailed discussion of a steady state and $V_{12}^{(\Lambda)}$, see App. B.

## 4.2 Ramsey contrast

We are interested in time evolution that follows an immersion of the impurity with $\mathbf{P} = 0$ in the bath. This corresponds to an abrupt quench of $V_{12}$ from zero to a finite value. For a non-dipolar impurity, this change can be realized using known experimental protocols, see, e.g, Ref. [10]. For a dipolar impurity, this could be done by first embedding the dipolar particle initially in the non-magnetic $m = 0$ state, at a magnetic field such that $a_{12} = 0$, and then transferring the impurity to the (maximally stretched) magnetic state of lowest Zeeman energy. This procedure would also result in this case to the desired abrupt switching-on of $V_{12}$.

To illustrate time evolution, we evaluate the contrast, i.e., the overlap between the polaron wave function and the initial state, $S(t) = \langle\psi(0)|\psi(t)\rangle$, where the initial state corresponds to a phononic vacuum and an impurity with $\mathbf{P} = 0$, i.e., $|\psi(0)\rangle = |0_\mathbf{k}, \mathbf{0}\rangle$. Note that $S(t)$ does not contain any information about the directional dependence of the dynamics. It is an experimentally relevant observable that contains only the information averaged over the sample. For the coherent variational ansatz, the contrast takes a simple form

$$S(t) = \exp\left[-i\phi(t) + \frac{1}{2}\sum_\mathbf{k}|\beta_\mathbf{k}(t)|^2\right]. \tag{3}$$

Using it and Eq. (2), we derive a semi-analytical expression:

$$S(t) = \exp\left[-\frac{iE_0 t}{\hbar} + \frac{4i}{3}\sqrt{\frac{na_{11}^3}{\pi}}\left(\frac{a_{12}}{a_{11}}\right)^2 \mathcal{R}\left(\frac{t}{t_n}\right)\right], \tag{4}$$

where the dipolar character of the problem is incorporated in $E_0$ and $\mathcal{R}$:

$$\mathcal{R}(x) = \int_0^1 du\, z(u)^2\left\{x\sqrt{w(u)} + e^{\frac{ixw(u)}{2} + \frac{i\pi}{4}}\sqrt{\frac{\pi x}{2}}(3i - w(u)x)\,\mathrm{erfc}\left[\frac{1+i}{2}\sqrt{xw(u)}\right]\right\},$$

and $E_0$ is the steady-state polaron energy, which can be conveniently written as $E_0 = g_{12}n + \gamma\int du\, z(u)^2 w(u)^{1/2}$ where $\gamma \sim (na_{11}^3)^{3/2}$; $z(u) = 1 + \frac{\sqrt{d_1 d_2}}{a_{12}}(3u^2 - 1)$ and $w(u) = 1 + \epsilon_{dd}(3u^2 - 1)$ are auxiliary functions, see App. B. The characteristic timescale $t_n = \frac{m}{8\pi n\hbar a_{11}}$ describes time evolution of a non-dipolar Bose gas at the mean-field level. As $t_n \sim \xi/c_0$, we interpret it as the timescale for a phonon to pass the region of the condensate distorted by the impurity. Note that it is natural to expect that $t_n$ defines polaron formation time, and as we show in the next subsection this is indeed the case. Indeed, assuming that the initial state contains excited phonons next to the impurity, polaron formation time is determined as the time needed for these phonons to leave the vicinity of the impurity, which is approximately $t_n$ [25,54]. For the considered parameters $t_n \sim 0.1$ms.

The short-time dynamics ($t \ll t_n$) of the contrast follows from Eq. (4)[3]

$$S(t) \approx 1 - (1 + i)\sqrt{\frac{t}{t_\Omega}} - i\frac{ng_{12}}{\hbar}t, \tag{5}$$

---

[3]Note that we disregard the term $i\frac{t}{t_\Omega}$, which is proportional to $V_{12}^4$ for consistency of our derivations. Note also that the quantum fluctuations term that enters the polaron energy, $\sim \gamma\int_0^1 du\, z(u)^2\sqrt{w(u)}t$, drops out of the expression. This term becomes relevant only at longer time scales.

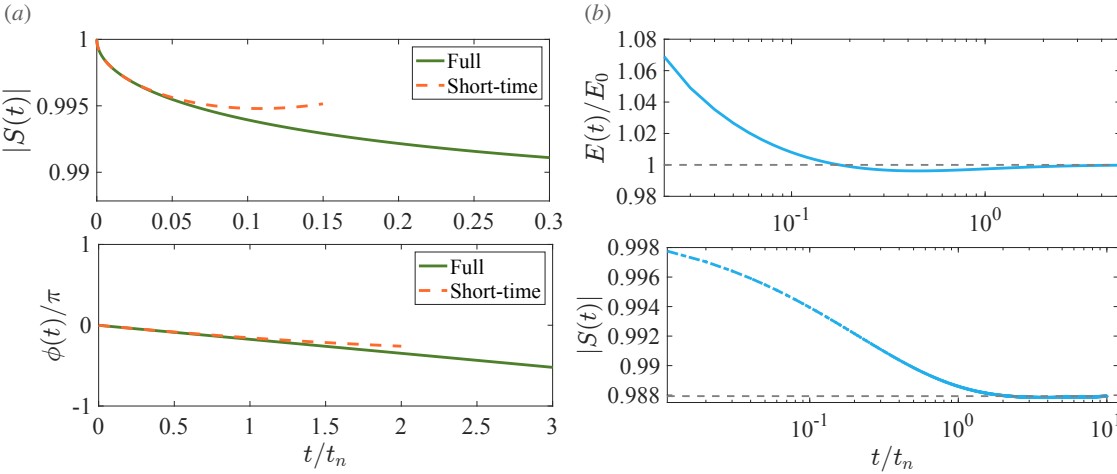

Figure 2: **Dipolar polaron contrast (a)** Absolute value and phase of the contrast (solid green curves) together with the corresponding short-time prediction of Eq. (5) (dashed orange curves) for a homogeneous dipolar quantum gas. **(b)** Instantaneous energy $E(t)$ and amplitude of the contrast as a function of time (solid blue curves); $E_0$ is the polaron energy in the steady state; $|S(t \to \infty)|$ converges to the quasiparticle residue $Z$ [Eq. (7)], which is shown with a dashed gray line. Note that $E(t)$ and $|S(t)|$ reach its asymptotic values on the timescales given by $t_n$. Our interpretation is that the quasiparticle (polaron) picture becomes valid after these timescales, i.e., the polaron is 'formed'. Here we used the gas parameter $na_{11}^3 = 5 \times 10^{-5}$; the coupling strengths are $d_1 = d_2 = 130a_0$, $\epsilon_{\mathrm{dd}} = 0.993$ and $a_{12} = a_{11}$. The time scales $t_n$ and $t_\Omega$ for these parameters and for Dy atoms are approximately 0.1ms and 0.2s.

where the initial dynamics is characterized by the timescale

$$t_\Omega = \frac{m}{32\pi\hbar n^2 \left(a_{12}^2 + \frac{4}{5}d_1 d_2\right)^2}, \tag{6}$$

which is determined only by low-energy two-body impurity-boson scattering and the density of the bosons, and does not contain any information about the boson-boson interactions. Our approach does not allow us to access the unitarity-limited universal dynamics [10, 55]. These dynamics can however be observed in current experiments only with strongly interacting gases making it irrelevant for our study.

At long times $t \gg t_n$, the phase and the absolute value of the contrast approach the steady-state quantities: the polaron energy $E_0$ and the quasiparticle residue, $Z = |S(t \to \infty)|$, which is given by

$$Z = \exp\left[-\frac{8}{3}\sqrt{\frac{na_{11}^3}{\pi}} \left(\frac{a_{12}}{a_{11}}\right)^2 \int_0^1 \frac{z(u)^2}{\sqrt{w(u)}}\mathrm{d}u\right]. \tag{7}$$

To illustrate the time evolution of the contrast for intermediate times, we shall first consider the system in a box potential, for which the density of the Bose gas is uniform.

## 4.3 Homogeneous case

We consider the scenario where $a_{11} \gtrsim d_1$, which corresponds to a stable system with dynamics strongly affected by dipole-dipole interactions. At $t < 0$, $V_{12} = 0$; at $t = 0$, the impurity-bath interaction is turned on, initiating the quench dynamics. Figure 2 (a) depicts the amplitude and

phase of the contrast as a function of time for a typical gas parameter $na_{11}^3 = 5 \times 10^{-5}$. Following Refs. [10, 56], we distinguish two regimes of the dynamics, namely, the two-body "relaxation" and the many-body "polaron formation", which for a non-dipolar impurity can be accessed experimentally by using interferometric techniques. During the initial "relaxation", which occurs for $t \ll t_n$, mainly two-body collisions between the impurity and the bath are relevant. Such two-body processes dominate the dynamics for times on the order of tenths of $\mu$s for typical experimental conditions, cf. the dashed curve in Fig. 2 (a).

In Fig. 2 (b), we present the time evolution of the contrast (4), and the instantaneous polaron energy, defined as $E(t) = \hbar\dot{\phi}(t)$. Time evolution of $E(t)$ reflects formation of excitations in the system via the following identity (see App. C):

$$E(t) = nV_{12}(\mathbf{0}) - \frac{1}{2}\sum_{\mathbf{k}}(\omega_{\mathbf{k}} + \epsilon_{\mathbf{k}})|\beta_{\mathbf{k}}|^2, \tag{8}$$

which connects $E(t)$ to the energy deposited to the medium and impurity states. The latter states are populated via recoil due to the conservation of momentum in phonon-impurity scattering. According to Eq. (8), $E(t = 0) \geq E(t)$ in agreement with Fig. 2 (b). In the "relaxation" regime, the energy of the system decreases as a function of time due to the energy exchange (dephasing) with the bosonic bath. For $t \sim t_n$, the system is in the "formation" region where collective excitations of the Bose gas are building up the quasiparticle. For $t \gg t_n$ the polaron is formed – its instantaneous energy is the steady-state energy, $E_0$; the absolute value of the contrast is the quasiparticle residue.

In Fig. 3 (a), we plot $|S(t)|$ as a function of time for different values of $\epsilon_{dd} < 1$ (recall that for $\epsilon_{dd} = 1$ the system is unstable against 3D collapse). Firstly, we consider the case $a_{11} = a_{12}$. We tune $\epsilon_{dd}$ by changing $a_{11}$ for fixed $d_1 = d_2$. For $a_{11} = a_{12}$ and $d_1 = d_2$, the boson-boson and boson-impurity interactions as well as the functions $z(u)$ and $\omega(u)$ are identical. This implies values of the residue close to unity even for $\epsilon_{dd} \to 1$, see Eq. (7). As $\epsilon_{dd}$ decreases, i.e., the bath becomes less dipolar, the quasiparticle residue also decreases. This happens due to the increase of $a_{11}$, which makes the bath more strongly interacting, rendering the dynamics *less* coherent.

In Fig. 3 (b), we present $|S(t)|$ for $a_{11} = 2a_{12}$. In this case, the residue vanishes in the limit $\epsilon_{dd} \to 1$ due to a slow logarithmic divergence of the integral in Eq. (7). Vanishing residue leads to a number of differences in comparison to the previously discussed symmetric case. In particular, for $\epsilon_{dd} \approx 1$, the absolute value of the contrast approaches the result of Eq. (7) at timescales much longer than the typical timescales of polaron formation, $t \simeq t_n$. By analyzing the function $R(t/t_n)$, we conclude that for $\epsilon_{dd} \approx 1$ long-time dynamics of the contrast is of the form

$$|S(t)| \simeq Z \exp\left[-\sqrt{\frac{na_{11}^3}{27\pi}}\left(\frac{4t_n(a_{12} - \sqrt{d_1 d_2})}{t(1 - \epsilon_{dd})a_{11}}\right)^2\right], \tag{9}$$

which allows us to introduce a relevant timescale close to the collapse: $t_\epsilon = t_n/(1 - \epsilon_{dd})$; Eq. (9) is valid for $t \gg t_\epsilon$ and $\sqrt{d_1 d_2} \neq \epsilon_{dd}a_{12}$. The timescale $t_\epsilon$ underlies the very slow dynamics of $|S(t)|$ and of the instantaneous polaron energy as $\epsilon_{dd} \to 1$. We illustrate this slowdown in the inset of Fig. 3 (b).

## 4.4 Anisotropy of time evolution

Up to now, we have considered quantities integrated over all directions, which cannot show the anisotropy of the time evolution. However, in general, the impurity experiences direction-dependent

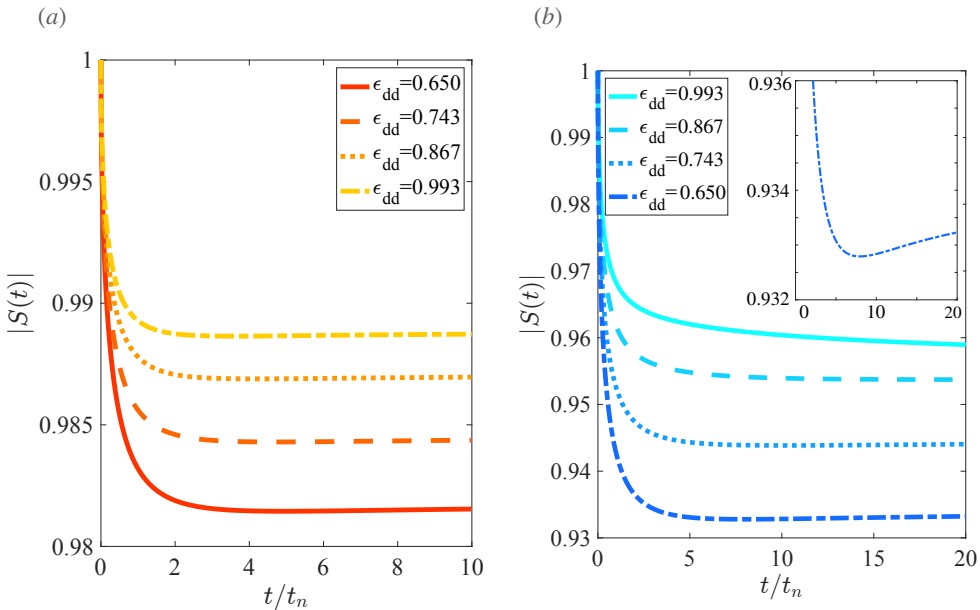

Figure 3: **Amplitude of the contrast** (a) Amplitude as a function of time for $a_{12} = a_{11}$ and different values of $\epsilon_{dd}$. (b) Amplitude as a function of time for $a_{12} = 2a_{11}$. In both plots, we fix the dipolar coupling strengths as $d_2 = d_1 = 130a_0$. (Inset) Slowdown of the dynamics due to a 'rich' dipolar character of the bath.

dynamics, with the limiting cases given when its momentum is parallel or perpendicular to the dipole direction; following Ref. [38] we call these the axial and radial cases, respectively. When $\epsilon_{dd} \neq 0$, the energy cost to create a phonon with $\theta = 0$ is higher than the energy cost to create a phonon with $\theta = \pi/2$, see Fig. 1 (a). Therefore, we expect that the movement of the impurity is affected more strongly in the radial direction (unless $V_{12}$ is also vanishing in this direction), which is further reflected in the values of the radial and axial effective masses in the steady state, see App. B. The latter effective mass is always weakly renormalized, while the former increases rapidly, implying slow diffusion in the radial direction.

To quantify the discussion above, and to illustrate the slowing-down of the radial dynamics compared to the axial dynamics, we study the number of excitations as a function of the angle: $B(\theta) = \sum_{\mathbf{k}'} |\beta_{\mathbf{k}'}|^2 \delta(\cos(\theta) - \cos(\theta'))$ where $\mathbf{k}' = \{k', \theta', \phi'\}$ is the integration index:

$$B_\theta(t) = \sqrt{\frac{16na_{11}^3}{\pi}} \left(\frac{a_{12}}{a_{11}}\right)^2 \int dx \frac{xz(u)^2 \left[1 - \cos\left(x^2 \frac{t}{t_n} + x\frac{t}{t_n}\sqrt{x^2 + w(u)}\right)\right]}{(x + \sqrt{x^2 + w(u)})^2 \sqrt{x^2 + w(u)}}. \tag{10}$$

This function features different timescales for different angles, see Fig. 1 (b), demonstrating that the time evolution of the dipolar polaron is drastically different from what is expected for short-range interactions. Physical insight into the anisotropy can be gained by considering the short-time dynamics, i.e., $t \ll t_n$. In this limit, we derive $B(\theta) \simeq \sqrt{t/t_F^\theta}$, where

$$t_F^\theta = \frac{m}{32\pi n^2 \left[a_{12} + \sqrt{d_1 d_2}(3\cos^2(\theta) - 1)\right]^4}. \tag{11}$$

The timescale $t_F^\theta$ depends on the angle, showing the difference in the dynamics parallel and perpendicular to the dipolar moment. One striking example is the dynamics with $a_{12} = \sqrt{d_1 d_2}$, which leads to weak interactions for $\theta = \pi/2$ and consequently $t_F^{\theta \to \pi/2} \to \infty$ [cf. the bottom panel of

Fig. 1 (b)].

Note that the anisotropy of $t_F^\theta$ is due to the dipolar nature of the impurity, and vanishes if we assume that $d_2 = 0$ [compare the two curves in the top panel of Fig. 1 (b) at $t \to 0$]. For longer times, the anisotropy of the dynamics is also driven by the anisotropy of the medium. It exists even if $d_2 = 0$, provided that $d_1 \neq 0$. This can be shown using $\beta_{\mathbf{k}}$ from the saddle point approximation. The existence of the anisotropy for $d_2 = 0$ can be also anticipated from the directional dependence of the effective mass, see App. B.

Finally, we mention another manifestation of anisotropy of time evolution that could be observed if the velocity of the impurity is close to the speed of sound. In this case, there is a "critical slowdown" dynamics (not to be confused with the slowdown presented in the inset of Fig. 3 (b) due to $\epsilon_{dd} \to 1$) [25]. In the non-dipolar case, the impurity exhibits a "critical slowdown" near the critical momentum $p_c = mc_0$. In the dipolar case, the speed of sound is direction dependent, hence, we expect that the "slowdown" occurs when the impurity momentum resonates with a specific sound mode in a particular direction. Therefore, the impurity will experience the "slowdown" more readily (i.e., for smaller values of the momentum of the impurity) in the radial direction – the effect will be particularly strong if $\epsilon_{dd}$ is close to one.

## 4.5 Trapped system

As we have shown, many properties of dipolar impurities in the homogeneous system can be understood (at least at the qualitative level) from the case with zero-range interactions by simply assuming that the $s$-wave scattering length depends on the direction. The physics of the problem becomes more intricate in the presence of inhomogeneous trapping, due to a possible interplay between the trap geometry and the anisotropy of the interaction. To illustrate this, we use the local-density approximation to study the experimentally relevant case of a system in a harmonic trap, see, e.g., [43]. We assume that the dipolar condensate is confined by the external potential $V(\rho, z) = \frac{1}{2}m\omega_z^2\left(\lambda^2\rho^2 + z^2\right)$, with dipoles pointing along the $z$-direction, $\lambda = \omega_\rho/\omega_z$, and study the role of different geometries on the dynamics of the contrast. For simplicity, we assume that the impurity does not experience any external potential.

In Fig. 4 (a) we present the amplitude and the phase of the contrast averaged over the Bose gas confined by a cigar-shaped ($\lambda = 0.1$), and a spherical ($\lambda = 1$) traps, which correspond to $(\omega_\rho, \omega_z) = 2\pi(50, 500)$Hz and $(\omega_\rho, \omega_z) = 2\pi(500, 500)$Hz, respectively. To calculate the contrast, we rely on the local-density approximation ($n \to n(\mathbf{r})$):

$$S = \frac{1}{N}\int \mathrm{d}^3\mathbf{r}\,n(\mathbf{r})\exp\left[-\frac{i\mathcal{E}(\mathbf{r})t}{\hbar} + \frac{4i}{3}\sqrt{\frac{n(\mathbf{r})a_{11}^3}{\pi}}\left(\frac{a_{12}}{a_{11}}\right)^2\mathcal{R}\left(\frac{8\pi\hbar a_{11}n(\mathbf{r})t}{m}\right)\right], \quad (12)$$

where $\mathcal{E}(\mathbf{r}) = \int d^3\mathbf{r}'V_{12}(\mathbf{r} - \mathbf{r}')n(\mathbf{r}) + \gamma(\mathbf{r})\int du z(u)^2 w(u)^{1/2}$. Note that we use the real-space representation of $V_{12}$ only at the mean-field level in $\mathcal{E}$. This allows us to account for the leading-order effect of the long-range tail of dipole-dipole interactions. The bosonic density for our system with $N = 2 \times 10^4$ atoms is conveniently calculated using the Gaussian variational ansatz [50]

$$n(\rho, z) = \frac{N}{\pi^{3/2}l_\rho^2 l_z a_H^3}\exp\left[-\frac{1}{a_H^2}\left(\frac{\rho^2}{l_\rho^2} + \frac{z^2}{l_z^2}\right)\right], \quad (13)$$

where $l_\rho$ and $l_z$ are variational parameters found by minimizing the energy; $a_H = \sqrt{\hbar/m\overline{\omega}}$ is the harmonic oscillator length with $\overline{\omega} = (\omega_\rho^2\omega_z)^{1/3}$. The Bose gas in a trap will have an anisotropic

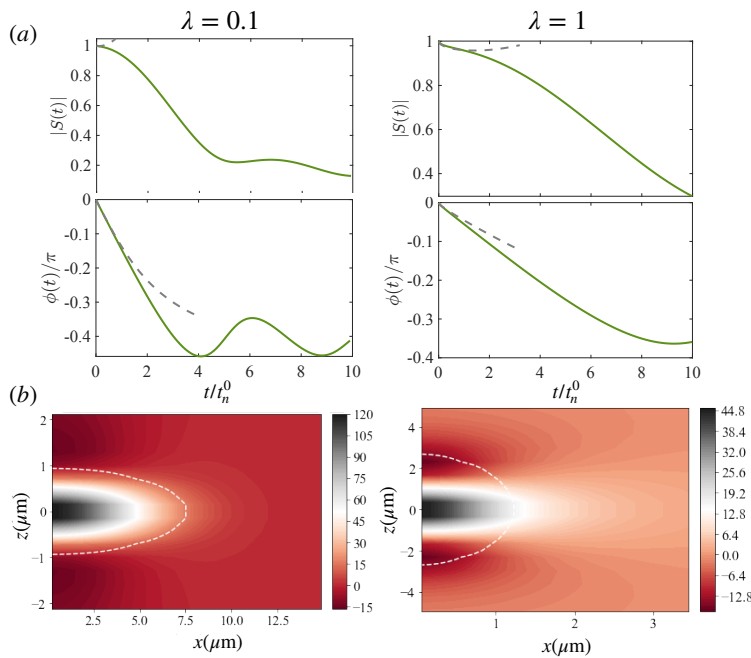

Figure 4: **Dynamics in a trap (a)** Contrast (amplitude and phase) as a function of time for an oblate trap $(\omega_\rho, \omega_z) = 2\pi(50, 500)$ (green solid), and for a spherical trap $(\omega_\rho, \omega_z) = 2\pi(500, 500)$ (grey dashed). Here, $\epsilon_{dd} \sim 1$ and $a_{12} = a_{11}$. **(b)** Steady-state local polaron energy (at the mean-field level) $E_0(\mathbf{r}) = \int d^3\mathbf{r}' V_{12}(\mathbf{r} - \mathbf{r}')n(\mathbf{r})$ for the two aforementioned cases. To illustrate the shape of the condensate, we draw (see dashed curves) a contour of the condensate when its density is at a 10% of the peak value. Here we use $a_{11} = 130a_0$ and $d_1 \approx d_2$

density, which plays an important role in time evolution, as can be seen by comparing the left and the right panels in Fig. 4(a).

The short-time dynamics of the contrast can be computed using Eq. (5)

$$S(t) \approx 1 - (1 - i)\beta\sqrt{\frac{t}{t_n^0}} - \frac{i}{4\sqrt{2}}\left[\frac{a_{12}}{a_{11}} + \frac{\sqrt{d_1 d_2}}{a_{11}}\kappa(\Delta^{-1})\right]\frac{t}{t_n^0}, \tag{14}$$

where $\beta = \sqrt{\frac{1}{2}n_0 a_{11}^3}\left(\frac{a_{12}^2 + \frac{4}{5}d_1 d_2}{a_{11}^2}\right)$ and the timescale depends on the peak density, $n_0 = n(0, 0)$, $t_n^0 = \frac{m}{8\pi\hbar n_0 a_{11}}$. The function $\kappa$ is defined as (cf. [50]) $\kappa(\Delta^{-1}) = \frac{(1 + 2\Delta^2)}{(\Delta^2 - 1)} - \frac{3\Delta^2 \arctan\sqrt{\Delta^2 - 1}}{(\Delta^2 - 1)^{3/2}}$ with $\Delta = l_z/l_\rho$. The last term in Eq. (14) originates from the local polaron energy at the level of first order perturbation theory (see App. E for more details). The interplay between the dipole-dipole interaction and the geometry of the potential enters Eq. (14) via the term $\sqrt{d_1 d_2}\kappa$. By changing the shape of the trap one varies $\kappa$ in the range $[-1, 2]$, and consequently the energy exchange between the impurity and the bath, see Fig. 4, which illustrates distinctly different approaches to the steady states for the two values of $\lambda$. Note that in the limiting case $a_{12} = -\sqrt{d_1 d_2}\kappa(l_\rho/l_z)$, the average polaron energy at the mean-field level vanishes, slowing down the dynamics. This regime amplifies the importance of the $\sqrt{t}$ term and beyond-mean-field effects (see Eq. (14)) in a dipolar mixture, suggesting the contrast as a probe of novel physics.

The local-density approximation utilized above is accurate only if the impurity can explore a small region of space. This assumption is adequate for studies of time evolution on timescales given by $t_n \sim 0.1$ms (compare with timescales for the dynamics in the trap $2\pi/\bar{\omega} \simeq 2$ms). Time

evolution of the impurity cloud at longer times deserves a further investigation. To motivate it, we present in Fig. 4(b) the local polaron energy $E_0(\mathbf{r})$ (see App. E) for $\lambda = 0.1$ and $\lambda = 1$. We see that in the former case the energy of the impurity is minimized when it is expelled from the condensate. In the latter case, the impurity prefers to localize outside the condensate along the $z$-axis (cf. Ref. [57]).

## 4.6 Summary and Outlook

We studied steady-state properties and quench dynamics of an impurity atom in a dipolar Bose-Einstein condensate. We solved Eq. (2) that determines a coherent-state variational ansatz employed to describe the system. Using our solution, we derived and analyzed a semi-analytical expression for the Ramsey contrast, see Eq. (4). In particular, we calculated short- and long-time dynamics of the contrast that elucidate initial few-body dynamics and approach of the system to the quasiparticle limit, see Eqs. (5) and (9). Furthermore, we proposed and illustrated a quantity suitable for studying anisotropic time evolution driven by dipole-dipole interactions, see Sec. 4.4. Finally, we employed a local-density approximation to analyze an experimentally relevant trapped system and showed that the dynamics of the system depends strongly on the shape of the trap, contrasting the system under consideration with cold-atom impurity systems where only contact interactions are important.

In this work the host medium is a Bose-Einstein condensate, however, our results can be easily extrapolated to impurities immersed in an isolated droplet or in a dipolar supersolid. A possible interesting future direction is to study the polaron dynamics across the superfluid-to-supersolid transition using dipolar or non-dipolar impurities embedded in dipolar media [58,59]. This transition is expected to have an unusual phase diagram [60], and the impurity dynamics may provide a tool to probe it.

Another direction is to consider multiple dipolar polarons and their interactions induced by the medium. In contrast to systems with short-range interactions [61–63], the anisotropic nature of the dipole-dipole potential will induce anisotropic correlations between impurities. These can be observed experimentally for example by studying the anisotropy of the impurity cloud in quench dynamics (cf. Ref. [64]).

Finally, we note that dipolar polarons in quantum gases may provide a valuable insight into out-of-equilibrium dynamics of indirect excitons modelled as electric dipoles. Interaction and trapping in these solid-state systems can be controlled, making it possible to study out-of-equilibrium dipolar excitons in atomically thin semiconductors [65]. These systems garner significant attention due to their unique optical properties, and their quantum simulation may not only deepen our understanding of dipolar excitations, but also lead to novel technological applications.

# Acknowledgements

We thank Lauriane Chomaz for useful discussions and comments on the manuscript. We also thank Ragheed Al Hyder for comments on the manuscript.

**Author contributions**   The project was conceived by L.A.P.A. All authors contributed to the analysis of numerical data, and to the development of the manuscript. A.G.V, G.B. and L.A.P.A. contributed equally to this work.

**Funding information**   G.B. acknowledges support from the Austrian Science Fund (FWF), under Project No. M2641-N27. This work is supported by the Deutsche Forschungsgemeinschaft (DFG,

German Research Foundation) under Germany's Excellence Strategy EXC2181/1-390900948 (the Heidelberg STRUCTURES Excellence Cluster). A. G. V. acknowledges support from the European Union's Horizon 2020 research and innovation programme under the Marie Skłodowska-Curie Grant Agreement No. 754411. L.A.P.A acknowledges by the PNRR MUR project PE0000023 - NQSTI and the Deutsche Forschungsgemeinschaft (DFG, German Research Foundation) under Germany's Excellence Strategy - EXC - 2123 Quantum Frontiers-390837967 and FOR2247.

## A  On the accuracy of the Fröhlich Hamiltonian

To investigate the effects of beyond-Fröhlich physics, let us consider the terms that describe other relevant scattering processes such as

$$\mathcal{H}_{\text{BF}} = \frac{1}{V}\sum_{\mathbf{k,q}} V^{(1)}_{\mathbf{kq}} e^{i(\mathbf{k-q})\cdot\hat{\mathbf{r}}} \hat{b}^{\dagger}_{\mathbf{k}}\hat{b}_{\mathbf{q}} + \frac{1}{V}\sum_{\mathbf{k,q}} V^{(2)}_{\mathbf{kq}} e^{i(\mathbf{k+q})\cdot\hat{\mathbf{r}}} \left(\hat{b}^{\dagger}_{\mathbf{k}}\hat{b}^{\dagger}_{\mathbf{q}} + \hat{b}_{-\mathbf{k}}\hat{b}_{-\mathbf{q}}\right), \quad \text{(A.1)}$$

where

$$V^{(1,2)}_{\mathbf{kk'}} = \frac{1}{2}V_{12}(\mathbf{k'}\mp\mathbf{k})\left[W_{\mathbf{k}}W_{\mathbf{k'}} \pm \frac{1}{W_{\mathbf{k}}W_{\mathbf{k'}}}\right]. \quad \text{(A.2)}$$

The upper (lower) sign corresponds to $V^{(1)}_{\mathbf{kk'}}$ ($V^{(2)}_{\mathbf{kk'}}$). This Hamiltonian leads to the following equations for the parameters $\beta_{\mathbf{k}}$, which appear in our variational ansatz,

$$i\hbar\dot{\beta}_{\mathbf{k}} = \frac{\sqrt{n}}{\sqrt{V}}V_{12}(\mathbf{k})W_{\mathbf{k}} + \Omega_{\mathbf{k}}\beta_{\mathbf{k}} + \frac{1}{V}\sum_{\mathbf{k'}} V^{(1)}_{\mathbf{kk'}}\beta_{\mathbf{k'}} + \frac{1}{V}\sum_{\mathbf{k'}} V^{(2)}_{\mathbf{kk'}}\beta^{*}_{\mathbf{k'}}. \quad \text{(A.3)}$$

The equation for $\phi(t)$ remain unchanged.

The last two terms in Eq. (A.3) describe the beyond-Fröhlich-Hamiltonian physics. Note that their effect is next-to-leading order (in powers of $V_{12}$) because the source term, $\sqrt{n}V_{12}(\mathbf{k})W_{\mathbf{k}}/\sqrt{V}$, for $\beta_{\mathbf{k}}$ is proportional to $\sqrt{n}V_{12}$, which is a small parameter for experimentally relevant densities. Therefore, these terms can be neglected in our work (at least within our variational approach), as the properties that we calculate are determined by $|\beta_{\mathbf{k}}|^2$. Note that the beyond-Fröhlich terms are important for the case $|a_{ij}| \gg d_i$, which can be described using frameworks developed to study non-dipolar polarons.

## B  Static properties of the system

Below, we provide four appendices with technical detials that support results presented in the main text. In App. B, we compute the polaron properties, including its energy, quasiparticle residue, and effective mass. The energy will later enter as an input in the calculation of the contrast. The energy is computed using second-order perturbation theory with either the bare impurity-boson perturbing potential or the renormalized one derived by excluding high-energy momentum states. In App. C we provide technical details for Eq. (2) of the main text. In App. D, we derive the analytical expression for the contrast, see Eq. (4) of the main text. In App. E, we assume a Gaussian density profile of the Bose gas to compute the energy in a trapped case. This allows us to highlight the underlying effects in polaron's dynamics in a trap.

**1. Lee-Low-Pines transformation (LLPt).**   In impurity problems, a transformation to the co-moving frame of the impurity is customary and simplifies the calculations. We transform the

Hamiltonian, $\mathcal{H}' \to \hat{S}^{-1}\mathcal{H}\hat{S}$, by means of the operator $\hat{S} = \exp\left(i\hat{\mathbf{r}} \cdot (\hat{\mathbf{P}} - \hat{\mathbf{P}}_{\mathrm{B}})\right)$ where $\hat{\mathbf{P}}$ is the total momentum of the system, and $\hat{\mathbf{P}}_{\mathrm{B}} = \hbar\sum_{\mathbf{k}} \mathbf{k}\hat{b}_{\mathbf{k}}^{\dagger}\hat{b}_{\mathbf{k}}$ is the momentum of the bosons. At the Fröhlich level, we derive

$$\hat{S}^{-1}\mathcal{H}\hat{S} = \frac{(\hat{\mathbf{P}} - \hat{\mathbf{P}}_{\mathrm{B}})^2}{2m} + \sum_{\mathbf{k}} \omega_{\mathbf{k}}\hat{b}_{\mathbf{k}}^{\dagger}\hat{b}_{\mathbf{k}} + nV_{12}(\mathbf{k}=0) + \frac{\sqrt{n}}{\sqrt{V}}\sum_{\mathbf{k}\neq\mathbf{0}} V_{12}(\mathbf{k})W_{\mathbf{k}}\left(\hat{b}_{\mathbf{k}} + \hat{b}_{-\mathbf{k}}^{\dagger}\right). \tag{B.1}$$

Note that the total momentum $\hat{\mathbf{P}}$ can be considered constant here.

**2. Ground-state energy.** The polaron energy is among the key quantities that describe static and dynamic properties of the system, in particular, the contrast. To use methods of many-body perturbation theory, we write the Hamiltonian from Eq. (B.1) as

$$\mathcal{H} = \mathcal{H}_0 + \mathcal{H}_{int},$$
$$\mathcal{H}_0 = \frac{(\mathbf{P} - \hat{\mathbf{P}}_{\mathrm{B}})^2}{2m} + \sum_{\mathbf{k}} \omega_{\mathbf{k}}\hat{b}_{\mathbf{k}}^{\dagger}\hat{b}_{\mathbf{k}}, \tag{B.2}$$
$$\mathcal{H}_{int} = nV_{12}(\mathbf{k}=0) + \frac{\sqrt{n}}{\sqrt{V}}\sum_{\mathbf{k}\neq\mathbf{0}} V_{12}(\mathbf{k})W_{\mathbf{k}}\left(\hat{b}_{\mathbf{k}} + \hat{b}_{-\mathbf{k}}^{\dagger}\right),$$

where $\mathbf{P}$ is a real vector – the total momentum of the system. The first-order correction in perturbation theory is trivial for the homogeneous case $E_0^{(1)} = \langle 0|\,\mathcal{H}_{int}\,|0\rangle$ where $|0\rangle$ is the state consisting of a single impurity with momentum $\mathbf{P}$ and the vacuum of phonons. The second-order correction reads

$$E_0^{(2)} = \sum_{\mathbf{k}\neq\mathbf{0}} \frac{|\langle\mathbf{k}|\,\mathcal{H}_{int}\,|0\rangle|^2}{E_0^{(0)} - E_{\mathbf{k}}^{(0)}} = -\frac{n}{V}\sum_{\mathbf{k}} \frac{\left|g_{12} + g_{12}^{(d)}\left(3\cos^2\theta_k - 1\right)\right|^2 W_{\mathbf{k}}^2}{\Omega_{\mathbf{k}}(\mathbf{P})}, \tag{B.3}$$

where $\Omega_{\mathbf{k}}(\mathbf{P}\to 0) \approx (\epsilon_{\mathbf{k}} + \omega_{\mathbf{k}})$ for slow impurities. Using the definition of the contact $g_{ij}$ and dipolar coupling strengths $g_{ij}^{(d)}$ defined in the mean text, we re-write this expression as

$$E_0^{(2)} = -\frac{32}{\sqrt{\pi}}(na_{11}^3)^{3/2}\frac{\hbar^2}{ma_{11}^2}\left(\frac{a_{12}}{a_{11}}\right)^2\int_{-1}^{1}\int_{0}^{\infty} du\,dx \frac{x^2 z(u)^2}{\sqrt{x^2 + w(u)}\left(x + \sqrt{x^2 + w(u)}\right)}. \tag{B.4}$$

where $x = k\xi/\sqrt{2}$, $\xi = \hbar/\sqrt{2mg_{11}n}$ and $u = \cos\theta_{\mathbf{k}}$. We have defined the auxiliary functions:

$$\begin{bmatrix} w(u) \\ z(u) \end{bmatrix} = \begin{bmatrix} 1 + \epsilon_{dd}\left(3u^2 - 1\right) \\ 1 + \bar{\epsilon}_{dd}\left(3u^2 - 1\right) \end{bmatrix} \tag{B.5}$$

with $\epsilon_{dd} = d_1/a_{11}$ and $\bar{\epsilon}_{dd} = \sqrt{d_1 d_2}/a_{12}$. The integral in Eq. (B.4) diverges and should be regularized. Similar to the contact case, we do so by disregarding the short-wavelength phonons which are irrelevant in the polaron physics. To this end, we note that the integral

$$\lim_{x\to\infty}\int_{-1}^{1}\int_{0}^{\infty} du\,dx \frac{x^2 z(u)^2}{\sqrt{x^2 + w(u)}\left(x + \sqrt{x^2 + w(u)}\right)} = \frac{1}{2}\int du\,z(u)^2. \tag{B.6}$$

should be removed from Eq. (B.4), which leads to

$$E_0^{(2)} = 32\sqrt{\pi}(na_{11}^3)^{3/2}\frac{\hbar^2}{ma_{11}^2}\left(\frac{a_{12}}{a_{11}}\right)^2\int_{-1}^{1} du\,dx\left\{\frac{z(u)^2}{2} - \frac{x^2 z(u)^2}{\sqrt{x^2 + w(u)}\left(x + \sqrt{x^2 + w(u)}\right)}\right\}. \tag{B.7}$$

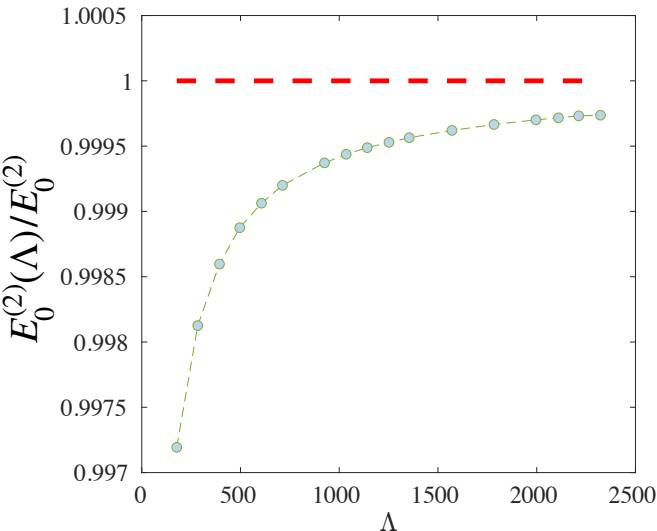

Figure S1: Beyond mean-field energy $E_0^{(2)}(\Lambda)/E_0^{(2)}$ computed with the exact Hugenholtz-Pines (HP) (red dashed line) formalism and the regularized correction (green points) as a function of the cut-off $\Lambda$ (in the units given by $n^{\frac{1}{3}}$). We find that $E_0^{(2)}(\Lambda) = E_{\rm HP}^{(2)} + \frac{\alpha}{\beta - \Lambda}$, confirming our regularization procedure. Here $a_{12} = 150a_0$ and $d_1 = d_2$.

This expression is further simplified

$$E_0^{(2)} = \frac{128\sqrt{\pi}}{3}(na_{11}^3)^{3/2}\frac{\hbar^2}{ma_{11}^2}\left(\frac{a_{12}}{a_{11}}\right)^2\int_0^1 du\, z(u)^2\sqrt{w(u)}, \tag{B.8}$$

which coincides with the expression obtained with the Hugenholtz-Pines formalism in [57].

Finally, we compare the mean-field energy to the derived correction $E_0^{(2)}$:

$$\frac{4\pi\hbar^2 a_{12}n}{mE_0^{(2)}} = \frac{3\sqrt{\pi}}{32\int_0^1 du\, z(u)^2\sqrt{w(u)}}\frac{a_{11}}{a_{12}}\frac{1}{\sqrt{na_{11}^3}}. \tag{B.9}$$

Assuming that $a_{11} = a_{12}$, $d_1 = d_2$, $na_{11}^3 \simeq 5 \times 10^{-5}$ and[4] $\epsilon_{dd} = 1$, we conclude that $\frac{4\pi\hbar^2 a_{12}n}{mE_0^{(2)}}$ is of the order of 10, validating the use of perturbation theory.

**3. Regularized potential.** In the previous section, we computed the polaron energy using bare potentials. Alternatively, we could have used a regularized potential for these calculations. To find forms of the regularized potentials, we note that a dipolar Bose gas within a mean-field approximation can be described with the potential from the first Born approximation, see, e.g., [66–68]. Therefore, as long as we focus on a weakly-interacting Bose gas, our expression for $V_{11}$ is accurate. For any beyond mean-field calculations, this potential clearly fails, as can be seen by solving a few-body problem already with $g_{ij}^d = 0$, see, e.g., Ref. [69]. Therefore, we need to suggest an effective description for $V_{12}$.

For our purposes, the effective interaction follows from the first and second terms in Born approximation. Using $V_{12} = \frac{4\pi\hbar^2}{m}\left(a_{12} + \sqrt{d_1 d_2}(3\cos^2\theta_{\mathbf{k}} - 1)\right)$ and

$$V_{12}^{(\Lambda)} = \frac{4\pi\hbar^2}{m}\left(a_{12} + \sqrt{d_1 d_2}\frac{8}{5\pi}\Lambda + \frac{2a_{12}^2}{\pi}\Lambda\right), \tag{B.10}$$

---

[4]Note that $E_0^{(2)}$ is well-behaved even for $\epsilon_{dd} = 1$ unlike the effective mass of the polaron, see below.

where $\Lambda$ is the high-momentum cut-off parameter, we write the equations for $\beta$ and $\phi$, regularized in the second-order in the impurity-boson interaction strength

$$i\hbar\dot{\beta}_{\mathbf{k}} = \frac{\sqrt{n}}{\sqrt{V}}V_{12}(\mathbf{k})W_{\mathbf{k}} + \Omega_{\mathbf{k}}\beta_{\mathbf{k}}, \tag{B.11}$$

$$\dot{\phi} = V_{12}^{\Lambda}n + \frac{\mathbf{P}^2 - \mathbf{P}_{\mathrm{B}}^2}{2m} + \frac{\sqrt{n}}{\sqrt{V}}\sum_{\mathbf{k}}W_{\mathbf{k}}V_{12}(\mathbf{k})\mathfrak{Re}[\beta_{\mathbf{k}}]. \tag{B.12}$$

Note that $V_{12}^{\Lambda}$ is used only in the equation for $\dot{\phi}$. This is expected – recall that the parameter $\beta_{\mathbf{k}}$ is proportional to $V_{12}$. Using a steady-state solution of these equations, we compute the polaron energy

$$E_0 = V_{12}^{\Lambda}n + \frac{\mathbf{P}^2}{2m} - \frac{n}{V}\sum_{\mathbf{k}}^{\Lambda}\frac{V_{12}^2(\mathbf{k})W_{\mathbf{k}}^2}{\omega_{\mathbf{k}} + \frac{\hbar^2\mathbf{k}^2}{2m} - \frac{\hbar\mathbf{k}\cdot\mathbf{P}}{m}}. \tag{B.13}$$

In the thermodynamic limit $\sum_{\mathbf{k}} \to V/(2\pi)^3\int d^3\mathbf{k}$ and for $\mathbf{P} = 0$, this expression reads

$$E_0 = V_{12}^{\Lambda}n - \int_0^{\frac{\Lambda}{n^{\frac{1}{3}}}}dk\int_{-1}^{1}dx\frac{8\hbar^2k^2n^{4/3}\left[a_{12} + \sqrt{d_1d_2}(3x^2 - 1)\right]^2}{m\sqrt{k^2 + 16\pi(a_{11} + d_1(3x^2 - 1))n^{1/3}}\left(k + \sqrt{k^2 + 16\pi(a_{11} + d_1(3x^2 - 1))n^{1/3}}\right)},$$

coinciding with the result within the Hugenholtz-Pines formalism (see Eq. (B.8)) in the limit $\Lambda \to \infty$. To show this, use the integral

$$\int_0^{\infty}dp\left(\frac{p^2}{\sqrt{p^2 + F}(p + \sqrt{p^2 + F})} - \frac{1}{2}\right) = -\frac{2\sqrt{F}}{3}, \tag{B.14}$$

where $F$ is some constant. In Fig. S1 we compare the polaron energy derived using the Hugenholtz-Pines formalism with the energy based upon the regularized potential.

**4. Quasiparticle residue.** At the level of second-order perturbation theory, the residue is

$$Z = 1 - \frac{1}{V}\sum_{\mathbf{k}}\frac{|\langle\mathbf{k}|\mathcal{H}_{int}|0\rangle|^2}{\left(E_0^{(0)} - E_{\mathbf{k}}^{(0)}\right)^2}, \tag{B.15}$$

which leads to

$$Z = 1 - \frac{8}{3\sqrt{\pi}}\left(\frac{a_{12}}{a_{11}}\right)^2\sqrt{na_{11}^3}\int_0^1 du\frac{z(u)^2}{w(u)^{1/2}}, \tag{B.16}$$

where (as before) $x = k\xi/\sqrt{2}$ and $u = \cos\theta_{\mathbf{k}}$. Note that this expression coincides with the first two terms of Taylor series of Eq. (7) of the main text, providing another validation of the variational ansatz.

**5. Effective mass.** The polaron energy as a function of $\mathbf{P}$ (see Eq. (B.13)) allows us to compute the effective masses of the polaron, $m_x, m_y$ and $m_z$, defined via the expression for $\mathbf{P} \to \mathbf{0}$

$$E_0(P) = E_0(\mathbf{P} = \mathbf{0}) + \frac{P_x^2}{2m_x} + \frac{P_y^2}{2m_y} + \frac{P_z^2}{2m_z}. \tag{B.17}$$

Assuming that the momentum of the impurity is much smaller that the speed of sound (in all directions), we derive

$$\frac{m}{m_{i=x,y,z}} = 1 - \frac{2\hbar^2n}{Vm}\sum_{\mathbf{k}}\frac{[V_{12}(\mathbf{k})W_{\mathbf{k}}]^2 k_i^2}{\left(\omega_{\mathbf{k}} + \frac{\hbar^2k^2}{2m}\right)^3}, \tag{B.18}$$

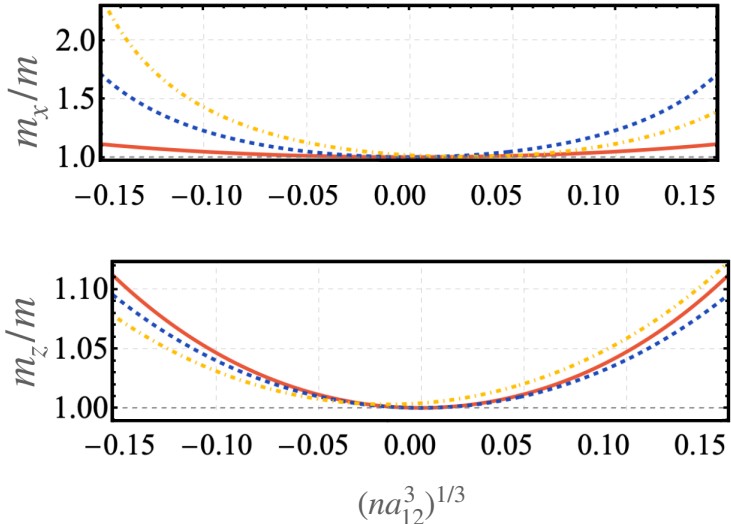

Figure S2: (Top) The effective mass along the $x$ axis, $m_x$, as a function of the (dimensionless) interspecies scattering length $a_{12}$, assuming that $a_{11} = 130.041a_0$. The different curves represent three different interaction regimes. The solid curve is for a system with only contact interactions, i.e., $d_1 = d_2 = 0$; the dashed curve illustrates the case with $d_2 = 0$; the dashed-dotted curve shows the mass for a dipolar impurity in a dipolar medium, see the text for details. (Bottom) The effective mass along the $z$ axis, $m_z$, as a function of the (dimensionless) interspecies scattering length $a_{12}$. The different curves represent three different regimes, see above and the text for details. Notice the different scaling of the $y$-axis in comparison to the top panel.

where (as in the main text) $m = m_2 = m_1$. Due to the axial symmetry of the problem, the directions $x$ and $y$ are identical, and therefore, we can focus only on the effective masses in $x$ and $z$ directions. These masses in the thermodynamic limit are

$$\frac{m}{m_x} = 1 - \frac{\hbar^2 n}{4\pi^3 m}\int d\mathbf{k}\frac{[V_{12}(\mathbf{k})W_{\mathbf{k}}]^2 k_x^2}{\left(\omega_{\mathbf{k}} + \frac{\hbar^2 k^2}{2m}\right)^3}, \qquad \frac{m}{m_z} = 1 - \frac{\hbar^2 n}{4\pi^3 m}\int d\mathbf{k}\frac{[V_{12}(\mathbf{k})W_{\mathbf{k}}]^2 k_z^2}{\left(\omega_{\mathbf{k}} + \frac{\hbar^2 k^2}{2m}\right)^3}. \quad \text{(B.19)}$$

Using the definitions from the discussion of the ground-state energy, we can also write the masses as

$$\begin{pmatrix} m/m_x \\ m/m_z \end{pmatrix} = 1 - \frac{8\sqrt{na_{11}^3}}{\sqrt{\pi}}\left(\frac{a_{12}}{a_{11}}\right)^2 \int dx du \frac{x^2 z(u)^2}{\sqrt{x^2 + w(u)}(x + \sqrt{x^2 + w(u)})^3}\begin{pmatrix} 1 - u^2 \\ 2u^2 \end{pmatrix}.$$

In the limit of $d_2 \to 0$ (non-dipolar) impurity these expressions reproduce the results presented in [38], providing us with another benchmark of our variational approach.

We identify three limiting cases: $d_1 = 0$ and arbitrary $d_2$; $d_1 \neq 0$ and $d_2 = 0$; $d_1 \neq 0$ and $d_2 \neq 0$. As expected, the effective masses $m_x$ and $m_z$ from Eq. (B.19) are identical if the medium is isotropic, i.e., $m_x = m_z$ if $d_1 = 0$. The difference between $m_x$ and $m_z$ for a non-dipolar impurity $d_2 \to 0$ ($a_{12}/d_2 \gg 1$) if $d_1 \neq 0$ has been discussed in Ref. [38]. For a dipolar gas and a dipolar impurity, our results show that if $\sqrt{d_1 d_2} \simeq a_{12}$ the effective masses are strongly affected by the dipole moments of both the impurity and bosons. To illustrate these limiting cases, we calculate the effective masses for Dysprosium atoms assuming $a_{11} = 130.041a_0$ and $n = 10^{20}m^{-3}$. The value of $a_{11}$ is chosen such that the system is stable, i.e., $V_{11}$ is always positive. In Fig. S2, we show the effective masses for (i) the contact case ($d_1 = d_2 = 0$); (ii) a dipolar medium with a contact impurity ($d_2 = 0, d_1 = 130a_0$); (iii) a dipolar medium with a dipolar impurity ($d_1 = d_2 = 130a_0$).

We see that the mass $m_z$ is only weakly affected by the dipolar nature of the interaction, whereas $m_x$ is strongly modified. This strong modification may be observed by means of momentum-resolved Bragg spectroscopy, thus, the effective mass anisotropy that we discuss here might be accessible within current experimental resolutions. Finally, let us analyze what happens in the limit $\epsilon_{dd} \to 1$, i.e., in the vicinity of the collapse of the Bose gas. The effective mass $m_z$ stays finite even if $\epsilon_{dd} = 1$. The mass $m_x$ however diverges. One can show that $m/m_x - 1 \sim \ln(1 - \epsilon_{dd})$.

Finally, we discuss the effect of self-localization (strong modification of the effective mass) that is predicted using the mean-field strong-coupling ansatz [70,71] (see also [72] for an attractive impurity in a trap) on our results. For a non-dipolar system, the defining dimensionless parameter for this localization is $\sqrt{16\pi n a_{12}^2}/\sqrt{a_{11}}$ [70], which is of the order of $0.1$ for the parameters used in our work. [Note that self-localization might occur only if this parameter is much larger than one.] Let us now speculate what happens for a dipolar system. To this end, we analyze a naive extension of the parameter above $\sqrt{16\pi n}[a_{12} + d_1(3\cos^2(\theta) - 1)]^2/\sqrt{a_{11} + d_1(3\cos^2(\theta) - 1)}$, where we assumed that $d_1 = d_2$. By estimating this parameter, we conclude that self-localization may occur only if $a_{12} \neq a_{11}$ and if $\epsilon_{dd} \simeq 0.999$ (for the considered values of the density and scattering lengths). Therefore, the self-localization effect does not affect the main findings of our study, and we leave its further investigation in dipolar systems for a future research. That research could rely on the mean-field ansatz in a co-moving frame [73–77], which (unlike the strong-coupling ansatz) allows one to study a strong modification of the density of the Bose gas preserving translational invariance of the problem.

## C   Derivation of Eq. (2)

For a non-dipolar case, Eq. (2) has been derived before [19]. For a dipolar case, the derivation is analogous. Here, we outline only a derivation of the equation for $\phi$, which follows from $\left\langle \psi(t) \left| i\hbar\partial_t - \hat{\mathcal{H}} \right| \psi(t) \right\rangle = 0$, see, e.g., Ref. [63].

First, we compute the expectation value of the time derivative

$$\langle \psi(t) | i\hbar\partial_t | \psi(t) \rangle = \hbar\dot{\phi} + i\hbar\sum \left[ \dot{\beta}_k(t)\beta_k^* - \dot{\beta}_k^*(t)\beta_k \right]/2, \tag{C.1}$$

where we have used the following properties of coherent states: $\langle \psi(t)|b_k|\psi(t)\rangle = \beta_k$ and $\langle \psi(t)|b_k^\dagger|\psi(t)\rangle = \beta_k^*$. Also it is convenient to use that $D(\alpha + \beta) = D(\alpha)D(\beta)e^{(\alpha\beta^* - \alpha^*\beta)/2}$, where $D(\beta) = e^{\beta b^\dagger - \beta^* b}$ when calculating the derivative of a coherent state.

Second, we compute the expectation value of the Hamiltonian

$$\langle \psi(t)|\hat{\mathcal{H}}|\psi(t)\rangle = \frac{(\mathbf{P} - \mathbf{P}_\mathrm{B})^2}{2m} + \sum_\mathbf{k}(\omega_\mathbf{k} + \epsilon_\mathbf{k})|\beta_\mathbf{k}|^2 + nV_{12}(\mathbf{0}) + 2\sqrt{\frac{n}{V}}\sum_\mathbf{k} V_{12}(\mathbf{k})W_\mathbf{k}\mathfrak{Re}[\beta_k], \tag{C.2}$$

where $\mathbf{P}_\mathrm{B} = \sum \mathbf{k}|\beta_\mathbf{k}|^2$; we used that $V_{12}(\mathbf{k})W_\mathbf{k} = V_{12}(-\mathbf{k})W_{-\mathbf{k}}$ and $D^\dagger(\beta)b^\dagger D(\beta) = b^\dagger + \beta^*$.

To proceed, we use the equation for $\beta_\mathbf{k}$ presented in the main text

$$i\hbar\dot{\beta}_\mathbf{k} = \sqrt{\frac{n}{V}}V_{12}(\mathbf{k})W_\mathbf{k} + \Omega_\mathbf{k}\beta_\mathbf{k} \tag{C.3}$$

to derive

$$\frac{i\hbar}{2}\sum_\mathbf{k} \left[ \dot{\beta}_k(t)\beta_k^* - \dot{\beta}_k^*(t)\beta_k \right] = \sum_\mathbf{k}\sqrt{\frac{n}{V}}V_{12}(\mathbf{k})W_\mathbf{k}\mathfrak{Re}[\beta_\mathbf{k}] + \sum_\mathbf{k}\Omega_\mathbf{k}|\beta_\mathbf{k}|^2. \tag{C.4}$$

With this expression, the equation for $\dot{\phi}$ reads

$$\hbar\dot{\phi} = \frac{(\mathbf{P} - \mathbf{P}_{\mathrm{B}})^2}{2m} + \sum_{\mathbf{k}} \left[ \frac{\hbar\mathbf{k}(\mathbf{P} - \mathbf{P}_{\mathrm{B}})}{m} \right] |\beta_{\mathbf{k}}|^2 + nV_{12}(\mathbf{0}) + \sqrt{\frac{n}{V}} \sum_{\mathbf{k} \neq 0} V_{12}(\mathbf{k})W_{\mathbf{k}}\mathfrak{Re}[\beta_k]. \quad \text{(C.5)}$$

This equation leads to Eq. (2) of the main text. Note that the terms proportional to $\mathbf{P}\mathbf{P}_{\mathrm{B}}$ vanish, and only the terms proportional to $\mathbf{P}_{\mathrm{B}}^2$ are relevant. However, these terms are the next-to-leading order and can be neglected in our calculations, as we discuss in the main text.

**Distribution of the energies in the system for $\mathbf{P} = 0$.** We can use Eq. (C.2) to understand the dynamics of the energy in the system initiated in the ground state, i.e., with $\mathbf{P} = 0$, which also implies that $\mathbf{P}_{\mathrm{B}} = 0$ as $\beta_{\mathbf{k}} = \beta_{-\mathbf{k}}$. First of all note that we have $\langle\psi(t)|\hat{\mathcal{H}}|\psi(t)\rangle = nV_{12}(\mathbf{0})$, i.e., the *total* energy is independent of time as it should. To show this conservation law, one can use the explicit solution to Eq. (C.3): $\beta_{\mathbf{k}}(t) = V_{12}(\mathbf{k})\frac{\sqrt{n}}{\sqrt{V}}\frac{W_{\mathbf{k}}}{\Omega_{\mathbf{k}}}\left[\exp\left(\frac{i\Omega_{\mathbf{k}}t}{\hbar}\right) - 1\right]$. The total energy is not to be confused with the energy of the polaron, $E(t)$, associated with $\hbar\dot{\phi}$. Its expression, $E(t) = nV_{12}(\mathbf{0}) - \frac{1}{2}\sum_{\mathbf{k}}(\hbar\omega_{\mathbf{k}} + \epsilon_{\mathbf{k}})|\beta_{\mathbf{k}}|^2$, was presented in the main text.

Let us provide some physical intuition for the terms in Eq. (C.2). The term $\sum_{\mathbf{k}}(\hbar\omega_{\mathbf{k}} + \epsilon_{\mathbf{k}})|\beta_{\mathbf{k}}|^2$ is the energy of excitations created during the dynamics. Note that a phonon is always accompanied by an excited state of an impurity. Indeed, recoil of the impurity is required by the conservation of the total momentum and explains the appearance of the combination $(\hbar\omega_{\mathbf{k}} + \epsilon_{\mathbf{k}})$ in the sum. The last term in Eq. (C.2) is always negative (or zero). It describes the energy stored in the boson-impurity interaction.

# D  Derivation of contrast

Here, we outline the most important steps in the derivation of Eq. (4) of the main text. Starting from Eq. (2), we write $\phi(t)$ as

$$\phi(t) = \frac{1}{\hbar}\int_0^t ds \left[ V_{12}(\mathbf{0})n + \frac{\sqrt{n}}{2\sqrt{V}}\sum_{\mathbf{k}} V_{12}(\mathbf{k})W_{\mathbf{k}}\left(\beta_{\mathbf{k}}(s) + \beta_{\mathbf{k}}^{\star}(s)\right) \right], \quad \text{(D.1)}$$

where the explicit solution of $\beta_{\mathbf{k}}(t)$ is $\beta_{\mathbf{k}}(t) = V_{12}(\mathbf{k})\frac{\sqrt{n}}{\sqrt{V}}\frac{W_{\mathbf{k}}}{\Omega_{\mathbf{k}}}\left[\exp\left(\frac{i\Omega_{\mathbf{k}}t}{\hbar}\right) - 1\right]$, which leads to

$$\phi(t) = \frac{1}{\hbar}\int_0^t ds \left\{ V_{12}(\mathbf{0})n + \frac{n}{V}\sum_{\mathbf{k}} \frac{V_{12}(\mathbf{k})^2 W_{\mathbf{k}}^2}{\Omega_{\mathbf{k}}}\left[\cos\left(\Omega_{\mathbf{k}}t/\hbar\right) - 1\right] \right\}. \quad \text{(D.2)}$$

It is convenient to split this integral into two parts with time-dependent and time-independent integrands:

$$\phi(t) = \overbrace{\int_0^t \frac{ds}{\hbar}\left\{ V_{12}(\mathbf{0})n - \frac{n}{V}\sum_{\mathbf{k}} \frac{V_{12}(\mathbf{k})^2 W_{\mathbf{k}}^2}{\Omega_{\mathbf{k}}} \right\}}^{I_1(t)} + \overbrace{\int_0^t \frac{ds}{\hbar}\left\{ \frac{n}{V}\sum_{\mathbf{k}} \frac{V_{12}(\mathbf{k})^2 W_{\mathbf{k}}^2}{\Omega_{\mathbf{k}}}\cos\left(\Omega_{\mathbf{k}}s/\hbar\right) \right\}}^{I_2(t)}.$$

The integral $I_1(t)$ is identical to the one appearing in our calculations of the polaron energy (see Sec. B), namely $I_1(t) = E_0 t/\hbar$. Note that regularization of the potential should be taken into account here. The second integral $I_2(t)$ can be written as

$$I_2(t) = \frac{1}{\hbar}\int_0^t ds \left\{ \frac{n}{V}\sum_{\mathbf{k}} \frac{V_{12}(\mathbf{k})^2 W_{\mathbf{k}}^2}{\Omega_{\mathbf{k}}}\cos\left(\Omega_{\mathbf{k}}s/\hbar\right) \right\} = \frac{n}{V}\sum_{\mathbf{k}} \frac{V_{12}(\mathbf{k})^2 W_{\mathbf{k}}^2}{\Omega_{\mathbf{k}}^2}\sin\left(\Omega_{\mathbf{k}}t/\hbar\right),$$

so that

$$\phi(t) = I_1(t) + I_2(t) = E_0 t/\hbar + \frac{n}{V}\sum_{\mathbf{k}} \frac{V_{12}(\mathbf{k})^2 W_{\mathbf{k}}^2}{\Omega_{\mathbf{k}}^2}\sin\left(\Omega_{\mathbf{k}} t/\hbar\right) \tag{D.3}$$

For the contrast, we also need to calculate the expression

$$\sum_{\mathbf{k}}|\beta_{\mathbf{k}}(t)|^2 = \frac{2n}{V}\sum_{\mathbf{k}}\frac{V_{12}(\mathbf{k})^2 W_{\mathbf{k}}^2}{\Omega_{\mathbf{k}}^2}\left[1 - \cos\left(\Omega_{\mathbf{k}} t/\hbar\right)\right]. \tag{D.4}$$

Using $\phi$ and $\beta$ in the definition of the contrast, $S(t) = \exp\left[-i\phi(t) + \frac{1}{2}\sum_{\mathbf{k}}|\beta_{\mathbf{k}}(t)|^2\right]$, we derive

$$S(t) = \exp\left[-i\bar{E}\bar{t} - \frac{4}{\sqrt{\pi}}\sqrt{na_{11}^3}\left(\frac{a_{12}}{a_{11}}\right)^2\int dudx \frac{x^3 z(u)^2}{\sqrt{x^2+w}\bar{\Omega}_x^2}\left(e^{-i\bar{\Omega}_x\bar{t}} - 1\right)\right], \tag{D.5}$$

where we use dimensionless variables, $\bar{\Omega}_x = \left(x^2 + x\sqrt{x^2+w}\right)$; $\bar{t} = t/t_n$ with $t_n = \frac{m}{8\pi\hbar na_{11}}$. For a non-dipolar condensate, $t_n$ is the time required for a phonon to travel through the region of the condensate distorted by the impurity. Indeed, $t_n \simeq \xi/c_0$, where the healing length $\xi$ defines the size of the distortion due the presence of the impurity. For a dipolar condensate, the anisotropy of time evolution is encoded in the prefactor $\bar{\Omega}_x$, which effectively leads to a direction-dependent speed of sound, $c(\theta)$.

The auxiliary functions $z(u)$ and $w(u)$ are defined in Eq. (B.5). The integrand appearing in Eq. (D.5) can be written as a primitive function and simplified further by using the procedure outlined in Appendix D of Ref. [25]. The primitive reads

$$K = -i\int duz(u)^2\int dx\int_0^{\bar{t}} ds\frac{x^3}{\sqrt{x^2+w(u)}\bar{\Omega}_x}e^{-i\bar{\Omega}_x s}. \tag{D.6}$$

Using $x = \bar{\Omega}_x/\sqrt{w(u) + 2\bar{\Omega}_x}$, $dx = (\bar{\Omega}_x + w)\left(w + 2\bar{\Omega}_x\right)^{-3/2} d\bar{\Omega}_x$ and $\sqrt{x^2 + w} = (\bar{\Omega}_x + w)(w + 2\bar{\Omega}_x)^{-1/2}$, the function $K$ is further simplified:

$$K = \int duz(u)^2\int d\bar{\Omega}_x\int ds\frac{\bar{\Omega}_x^2}{\left(w + 2\bar{\Omega}_x\right)^{5/2}}e^{-i\bar{\Omega}_x s}. \tag{D.7}$$

In this expression, the integrals over $\bar{\Omega}_x$ and $s$ can be computed:

$$\int_0^\infty d\bar{\Omega}_x\int_0^{\bar{t}} ds\frac{\bar{\Omega}_x^2 e^{-i\bar{\Omega}_x s}}{(w + 2\bar{\Omega}_x)^{5/2}} = \tag{D.8}$$
$$\left\{-\bar{t}\sqrt{w(u)} + \frac{1+i}{2}\exp\left[\frac{i}{2}w(u)\bar{t}\right]\sqrt{\pi\bar{t}}\left(3i - w(u)\bar{t}\right)\mathrm{erfc}\left[\frac{1+i}{2}\sqrt{\bar{t}w(u)}\right]\right\},$$

leading to the function $\mathcal{R}\left(\bar{t}\right)$ with $\bar{t} = t/t_n$ from the main text.

# E Energy for a system in a trap

We compute the contrast for an inhomogeneous density via

$$S(t) = \frac{1}{N}\int d^3\mathbf{r}S(\mathbf{r}, t)n(\mathbf{r}). \tag{E.1}$$

Note that in the homogeneous case the dipolar mean-field energy is zero, but for the trapped case, it does not vanish. In order to evaluate it, we compute

$$E_0(\boldsymbol{r}) = \int d^3\mathbf{r}' V_{12}(\mathbf{r} - \mathbf{r}') n(\mathbf{r}'), \tag{E.2}$$

where $V(\mathbf{r} - \mathbf{r}') = \frac{g_{12}^d}{4\pi|\mathbf{r}-\mathbf{r}'|^3}\left(\frac{1}{3} - \cos^2\theta\right)$ is the DDI interaction in real space. To compute the energy, we employ the convolution theorem,

$$E_0(\boldsymbol{r}) = \int d^3\mathbf{r}' V_{12}(\mathbf{r} - \mathbf{r}') n(\mathbf{r}) = \mathcal{F}^{-1}\left\{\mathcal{F}[V](\mathbf{k})\mathcal{F}[n](\mathbf{k})\right\},$$

where $\mathcal{F}$ and $\mathcal{F}^{-1}$ denote Fourier and Inverse Fourier transforms, respectively. The Fourier transform of the DDI interaction is $\mathcal{F}[V_{12}](\mathbf{k}) = g_{12}^{(d)}\left(3\cos^2\theta_{\mathbf{k}} - 1\right)$.

Let us assume a Gaussian density profile of a trapped condensate:

$$n(\rho, z) = \frac{N}{\pi^{3/2} l_\rho^2 l_z a_H^3} \exp\left[-\frac{1}{a_H^2}\left(\frac{\rho^2}{l_\rho^2} + \frac{z^2}{l_z^2}\right)\right], \tag{E.3}$$

where $a_H = \sqrt{\hbar/m\overline{\omega}}$ is the harmonic oscillator length, $\overline{\omega} = (\omega_\rho^2\omega_z)^{1/3}$; $l_\rho$ and $l_z$ are variational parameters that depend on the aspect ratio $\lambda = \omega_\rho/\omega_z$. Using that $\mathcal{F}[n](\mathbf{k}) = N\exp\left[-\frac{a_H}{4}\left(k_\rho^2 l_\rho^2 + k_z^2 l_z^2\right)\right]$, we write

$$E_0(\boldsymbol{r}) = \mathcal{F}^{-1}\left[\mathcal{F}^{-1}(V)\mathcal{F}[n](\mathbf{k})\right] =$$
$$g_{12}^d N \int \frac{d^3\mathbf{k}}{(2\pi)^3}\left[\frac{3k_z^2}{k_\rho^2 + k_z^2} - 1\right]\exp\left[-\frac{a_H^4}{4}\left(k_\rho^2 l_\rho^2 + k_z^2 l_z^2\right)\right]\exp\left[-j\mathbf{k}\cdot\mathbf{r}\right], \tag{E.4}$$

or (in dimensionless units):

$$E_0(\boldsymbol{r}) = \frac{g_{12}^d N}{l_\rho^2 l_z a_H^3}\int \frac{d^3\mathbf{q}}{(2\pi)^3}\left[\frac{3q_z^2}{\Lambda^2 q_\rho^2 + q_z^2} - 1\right]\exp\left[-\frac{1}{4}\left(q_\rho^2 + q_z^2\right)\right]\exp\left[-j\mathbf{q}\cdot\tilde{\mathbf{r}}\right], \tag{E.5}$$

where $\tilde{\mathbf{r}} = (\tilde{x}, \tilde{y}, \tilde{z}) = \frac{1}{a_H}\left(\frac{x}{l_\rho}, \frac{y}{l_\rho}, \frac{z}{l_z}\right)$ and $\Lambda = l_z/l_\rho$. In spherical coordinates, we write this expression as

$$E_0(\boldsymbol{r}) = \frac{1}{(2\pi)^3}g_{12}^d N \frac{1}{l_\rho^2 l_z a_H^3}\int dq q^2 \int \sin\theta_q\left[\frac{3\cos^2\theta_q}{\Lambda^2 \sin\theta_q^2 + \cos^2\theta_q} - 1\right]\exp\left[-\frac{1}{4}q^2\right]$$

$$\overbrace{}^{2\pi J_0\left[q\sin\theta_q\sqrt{\tilde{x}^2+\tilde{y}^2}\right]} \tag{E.6}$$

$$\times \exp\left[-jq\cos\theta_q\tilde{z}\right]\overbrace{\int_0^{2\pi} d\phi \exp\left[-jq\sin\theta_q\left(\tilde{x}\cos\phi + \tilde{y}\sin\phi\right)\right]},$$

where the integral in $\phi$ produces a Bessel function $J_0$. Using a variable $u = \cos\theta_q$, we write

$$E_0(\boldsymbol{r}) = \frac{g_{12}^d N}{l_\rho^2 l_z a_H^3}\int \frac{dq}{4\pi^2}\int du q^2\left[\frac{3u^2}{\Lambda^2\left(1 - u^2\right) + u^2} - 1\right]J_0\left[q\sqrt{1 - u^2}\sqrt{\tilde{x}^2 + \tilde{y}^2}\right]e^{-\frac{1}{4}q^2 - jqu\tilde{z}}.$$

We write this expression as

$$E_0(\boldsymbol{\rho}, z) = g_{12}^d n_0 \Omega(\boldsymbol{\rho}, z), \qquad \text{where} \qquad \Omega(\boldsymbol{\rho}, z) = \int dq \int du f(q, u, \boldsymbol{\rho}, z),$$

$$f(q, u, \boldsymbol{\rho}, z) = \frac{1}{4\sqrt{\pi}}q^2\left[\frac{3u^2}{\Lambda^2\left(1 - u^2\right) + u^2} - 1\right]J_0\left[q\sqrt{1 - u^2}\tilde{\rho}\right]e^{-\frac{1}{4}q^2 - jqu\tilde{z}}.$$

The polaron energy can be conveniently written in the units of energy $E_n = \hbar/t_n = 8\pi\hbar^2 n_0 a_{11}/m$ as

$$\frac{E_0(\boldsymbol{\rho}, z)}{E_n} = \frac{1}{2}\frac{\sqrt{d_1 d_2}}{a_{11}}\Omega(\boldsymbol{\rho}, z). \tag{E.7}$$

This quantity is illustrated in Fig. 4 of the main text.

The spatial integral over the energy, $\epsilon = \frac{1}{N}\int d^3\mathbf{r} g_{12}^d n_0 \Omega(\boldsymbol{\rho}, z) n(\mathbf{r})$, is known (cf. Ref. [50]). It reads

$$\epsilon = \frac{1}{N}\int d^3\mathbf{r}\int d^3\mathbf{r}' V_{12}(\mathbf{r} - \mathbf{r}')n(\mathbf{r}')n(\mathbf{r}) = \frac{1}{2\sqrt{2}}n_0 g_{12}^d \kappa(\Lambda^{-1}), \tag{E.8}$$

where

$$\kappa(\Lambda^{-1}) = \frac{(1 + 2\Lambda^2)}{(\Lambda^2 - 1)} - \frac{3\Lambda^2 \arctan\sqrt{\Lambda^2 - 1}}{(\Lambda^2 - 1)^{3/2}}.$$

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
