# Peer review of "Non-equilibrium dynamics of dipolar polarons"

_SciPost Physics_

## Round 1 · Referee Report · Anonymous (Referee 2) · 2023-7-19

Strengths

1- Very interesting topic and an original contribution
2- Well-written manuscript
3- Some interesting results clearly presented by the figures and conclusions.

Weaknesses

1- Some details of the argument and derivation might need to be included.

Report

Review Report for the Manuscript "Non-equilibrium dynamics of dipolar polarons"

This manuscript presents a perturbation study of the polaron physics of a weakly-interacting impurity immersed in a dipolar Bose-Einstein condensate. The authors employ the time-dependent variational theory on the polaron Hamiltonian in the moving frame of the polaron, obtained through the Lee-Low-Pines transformation. The paper explores various aspects of the polaron, such as steady-state energy, residue, effective mass, and relaxation dynamics. Additionally, the authors investigate the trapped system using a local-density approximation. Overall, the topic is interesting, and the manuscript is well-written and provides clear motivations, explanations, and detailed analysis. However, there are some concerns regarding the formalism that should be addressed before making a final recommendation.

(1) Derivation of Eqs. (2):
It would be beneficial to include a more detailed derivation of Eqs. (2) from the time-dependent variational ansatz that minimizes the action functional. Specifically, it would be helpful to explain why the second term on the right-hand side of the second equation in Eqs. (2) is not expressed as (P - P_B)^2/2m, given the naïve expectation that it should resemble the Hamiltonian in Eq. (S4).

(2) Neglect of P_B in the perturbation analysis:
The authors state that in the Lee-Low-Pines transformation, P_B (which is related to the recoil of the impurity by the phonons) is neglected in their perturbation analysis because it only enters in the next-to-leading order. However, neglecting P_B implies an effective Hamiltonian equivalent to that of an infinitely heavy impurity. This contradicts the assumption of equal masses for the impurity and background bosons, where recoil should play a significant role. It is necessary to provide a more detailed discussion justifying the neglect of P_B and its underlying physical picture.

(3) Self-trapping effect and earlier works:
Considering that the authors focus on weakly-interacting impurities, it would be valuable to mention early works on ultracold polarons that consider the self-trapping effect. The self-trapping effect leads to the spontaneous breaking of spatial translational symmetry, and the momentum P is no longer a good quantum number. Although the mean-field description fails for strongly interacting impurities, it is likely that the self-trapping effect remains relevant for weakly interacting impurities, at least for static state analysis. (I would expect that for dynamical analysis, this is less relevant, as the weakly-interacting BEC is a slowly-responded fluid.) This effect becomes particularly interesting in systems with long-range interactions. It would be appropriate to reference earlier works such as [Phys. Rev. Lett. 96, 210401 (2006), Phys. Rev. A 73, 043608 (2006)] and relevant literature in this context.

Some Minor Suggestions:

1. Discuss the underlying physics of relaxation:
Since there are no decay channels mentioned, it would be helpful for the authors to comment on the underlying physics of the relaxation process. It could be speculated that the relaxation is caused by some dephasing process. Providing insights into this aspect would enhance the understanding of the dynamics studied.

2. Perturbation argument in droplet or supersolid backgrounds:
While the authors claim that the results can be extrapolated easily to droplet or supersolid backgrounds, it is important to note that the density in such regimes might be significantly higher than in a Bose-Einstein condensate. Consequently, the validity of the perturbation argument may no longer hold. It would be fair to mention this limitation to the readers in the Outlook section.

Overall, the manuscript is well-written and addresses an interesting topic. By addressing the concerns raised and incorporating the suggestions provided, the authors can further improve the paper and make it suitable for publication.

---

## Round 1 · Referee Report · Anonymous (Referee 3) · 2023-8-21

Strengths

1) Interesting open concept
2) Analytical estimates
3) Original results

Weaknesses

1) Further argumentation of specific descriptions is needed

Report

In this work, the authors study the non-equilibrium dynamics of dipolar Bose polarons after an interaction quench of the impurity-boson coupling. Certain dynamical features of this system are calculated using a time-dependent variational approach and exploiting the Lee-Low-Pines transformation which allows also for analytical estimates and for distinguishing the relevant timescales. They extract several experimentally relevant observables such as the Ramsey contrast and quasiparticle effective mass. The main result is the manifestation of the polaron anisotropic relaxation dynamics due to the dipolar interactions. Trap effects are also explored within the local density approximation.

I find these results interesting towards the understanding of polarons in anisotropic media. Moreover, the submission is timely and the setup will be accessible in near future cold atom experiments. The manuscript is also well written. However, I have some comments and questions regarding the presentation and the interpretation of the presented results. If the authors are able to address these concerns which I provide below, the work will be further advanced and be suitable for publication in SciPost Physics. Suggestions for improvement follow

1) In the introduction, the statement “collective excitations build the subsequent time evolution” is not clear. What type of collective excitations are meant and in which context? I guess the few-body correlations should also play a role at long evolution times invalidating the mean-field approximation? Please elaborate at least briefly on this point.

2) In the introduction the term short-time polaron physics is stated. This becomes particularly confusing since a sentence later the term initial short time dynamics is used. Please provide an order of magnitude of what is actually meant.

3) In the same paragraph, it would be beneficial for the reader to already explain in a sentence the context of the Ramsey contrast. Otherwise, the comment provided at the end of the Introduction should be moved to this point and explain which interactions are meant in the overlap.

4) Since the authors provide a list of possible techniques that have been used to describe polaron physics it would be fair to also comment on other variational methods that have been extensively employed.

5) I would also encourage the authors to describe their main specific results in a more clear manner in the Introduction.

6) I wonder whether in the single impurity limit that is considered in Section 2, dipole-dipole interactions between the impurity and the bath are indeed experimentally justified. Please comment.

7) The fact that the relaxation timescales are less for a head-to-to-tail arrangement should be explained. Is it related to the attractive nature of the interaction potential and how? Please comment. The term “relaxation time” should also be explicitly explained since it has different interpretations in the literature.

8) It would be interesting to comment whether quantum fluctuations (in terms of second order perturbation) play any role in the static (e.g. ground state) polaron energy for the considered interactions. In other words, are there any deviations from the mean field contribution?

9) In the contrast defined in Equation (3), I guess the initial wave function refers to the situation where all interactions are switched-off. Is this correct? Please comment explicitly.

10) What is the physical context and the formula of the timescale t_n appearing in Equation (4)? How is it compared to t_\Omega?

11) In dipolar systems it is known that loss channels such as three-body recombination play an important role. Can the authors comment on the effect of such processes in their system.

12) The decay of the contrast shown in Figure is relatively weak. Would it be possible to be observed in the experiment? Can the authors provide their opinion on this issue?

13) Does the system exhibits any universal characteristics for long evolution times where e.g. the contract is saturated?

14) I expect that depending on the \epsilon_dd parameter the phase of the bath is modified. How is this fact be reflected in the contrast?

15) I would expect that the diffusion process is accompanied by energy redistribution between the impurity and the medium. Can the authors comment on this issue?

16) Does the fact that the impurity slowdown occurs faster for \epsilon_dd ->1 is related to the superfluid character of the medium or not? Please clarify.

17) I am not sure that the trap frequencies provided before Equation (11) have been used in dipolar condensates. If they have been used please provide relevant references.

18) In the outlook section I would suggest to add at least a brief description of the new results presented in the main text. This will certainly facilitate the understanding of the reader.

---

## Editorial Decision

resubmitted